# COVID-19 Worries and Insomnia: A Follow-Up Study

**DOI:** 10.3390/ijerph20054568

**Published:** 2023-03-04

**Authors:** Lily A. Brown, Yiqin Zhu, Gabriella E. Hamlett, Tyler M. Moore, Grace E. DiDomenico, Elina Visoki, David M. Greenberg, Ruben C. Gur, Raquel E. Gur, Ran Barzilay

**Affiliations:** 1Department of Psychiatry, School of Medicine, University of Pennsylvania Perelman, 3535 Market Street Suite 600N, Philadelphia, PA 19104, USA; 2Lifespan Brain Institute of the Children’s Hospital of Philadelphia and Penn Medicine, Philadelphia, PA 19104, USA; 3Department of Music, Bar Ilan University, Ramat Gan 5290002, Israel; 4Interdisciplinary Department of Social Sciences, Bar-Ilan University, Ramat Gan 5290002, Israel; 5Autism Research Centre, Department of Psychiatry, University of Cambridge, Cambridge CB2 8AH, UK; 6Department of Child Adolescent Psychiatry and Behavioral Sciences, Children’s Hospital of Philadelphia, Philadelphia, PA 19104, USA

**Keywords:** worry, insomnia, COVID-19 pandemic, anxiety

## Abstract

The COVID-19 pandemic was associated with significant increases in sleep disorder symptoms and chronic worry. We previously demonstrated that worry about the pandemic was more strongly associated with subsequent insomnia than the converse during the acute (first 6 months) phase of the pandemic. In this report, we evaluated whether that association held over one year of the pandemic. Participants (*n* = 3560) completed self-reported surveys of worries about the pandemic, exposure to virus risk factors, and the Insomnia Severity Index on five occasions throughout the course of one year. In cross-sectional analyses, insomnia was more consistently associated with worries about the pandemic than exposure to COVID-19 risk factors. In mixed-effects models, changes in worries predicted changes in insomnia and vice versa. This bidirectional relationship was further confirmed in cross-lagged panel models. Clinically, these findings suggest that during a global disaster, patients who report elevations in either worry or insomnia should be considered for evidence-based treatments for these symptoms to prevent secondary symptoms in the future. Future research should evaluate the extent to which dissemination of evidence-based practices for chronic worry (a core feature of generalized anxiety disorder or illness anxiety disorder) or insomnia reduces the development of co-occurring symptoms during a global disaster.

## 1. Worry about COVID-19 and Insomnia: A Follow-Up Study

The coronavirus (COVID-19) global pandemic was associated with exacerbations in insomnia and anxiety. The prevalence of insomnia increased from 17.6% to 19.0% in a national sample of US college students [1], while increased from 7.4% [2] to 14% [3] in the general population in Israel. The average nurse met the criteria for subthreshold insomnia even 18 months into the pandemic (with 41% reporting moderate to severe insomnia) [4]. Among individuals employed in critical societal occupations (healthcare, education, emergency services, etc.), and in the general population, insomnia severity rates during the pandemic were significantly higher compared to normative pre-pandemic rates [5,6,7]. Likewise, rates of chronic worry, most commonly documented as generalized anxiety disorder (GAD), skyrocketed during the pandemic, with prevalence estimates as high as 44% [8]. Moreover, longitudinal studies on prevalence rates of mental health diagnoses generally demonstrated significant worsening over time during the pandemic [9].

Our team previously reported on the longitudinal associations between worry and insomnia over the first six months of the pandemic in a large sample of participants who were recruited from the United States and Israel [10]. This is an important topic because of the exacerbations in worry and insomnia during the pandemic, yet no researchers prior to our team explored the temporal direction of worry and insomnia during the acute phase of the global disaster. First, we found that insomnia severity was more reliably associated with worry about COVID-19 than exposure to COVID-19 risk factors (e.g., testing positive for COVID-19, knowing someone who had COVID-19, etc.), in cross-sectional analyses. Second, in mixed-effects models exploring the change over time in worry as a predictor of the change over time in insomnia (and vice versa), we found evidence supporting both directions. To further understand the temporal nature of these effects, we conducted cross-lagged panel analyses which revealed that during the acute phase of the pandemic (6 months after the global declaration of the pandemic) worry about the pandemic was a stronger predictor of subsequent insomnia than the converse. However, as mentioned above, prevalence rates of mental health diagnoses only increased as the pandemic continued, and it is important to understand whether the effects between worry and insomnia held after the acute phase was completed.

In the current study, we continued observations of participants for two additional time points (five total observations) over the course of one year of the pandemic. We followed an identical analytic approach in this report as in our original report [10] and anticipated that the temporal direction of effects that were observed in the acute phase of the pandemic would hold or become even stronger as the pandemic continued. Specifically, we hypothesized that insomnia would be more strongly associated with worries about COVID-19 than actual exposure to COVID-19 risk factors. Second, we hypothesized that mixed-effects analyses would demonstrate that change over time in worry was associated with change over time in insomnia and vice versa. Third, when conducting a more rigorous evaluation of directionality in cross-lagged panel analyses, we hypothesized that worry about COVID-19 would remain a stronger predictor of insomnia than the converse.

## 2. Methods

### 2.1. Participants

Participants (demographics in Table 1) completed online surveys in English or Hebrew through a crowdsourcing website (www.covid19resilience.org, accessed on 5 December 2022) [11]. The study was advertised through: (1) the researchers’ social networks, including emails to colleagues around the world; (2) social media; (3) the University of Pennsylvania and Children’s Hospital of Philadelphia internal notifications and websites; and (4) organizational mailing lists. There were no inclusion or exclusion criteria for participant selection. More details, especially the distribution of self-reported history of diagnosis, are reported elsewhere [10]. Of participants available for inclusion in our initial report (who could then be followed for additional observations in the current report), 7690 participants started the survey and 3560 (46.3%) provided data on the Insomnia Severity Index at Time 1.

### 2.2. Measures

The Insomnia Severity Index (ISI) [12]. We used ISI to measure participants’ insomnia symptoms over the prior two weeks. Participants were asked to rate on a 5 Likert scale ranging from “no problems” to “very severe”. The ISI is a reliable and valid instrument to quantify perceived insomnia severity [12]. Sum scores of the seven items (range: 0–28) were calculated to indicate insomnia severity, and higher scores indicate more symptom severity. 

COVID-19-related worries. We used six items to measure worries about COVID-19, including (1) contracting COVID-19; (2) dying from COVID-19; (3) family members contracting COVID-19; (4) unknowingly infecting others with COVID-19; (5) currently having COVID-19; and (6) having significant financial burden because of the COVID-19 pandemic. Participants were asked to rate the degree to which they were worried about various COVID-19-related outcomes on a 5 Likert scale ranging from “not at all” to “a great deal”. 

COVID-19-related exposures. We used five dichotomous items to measure life events about COVID-19, including (1) being tested for COVID-19; (2) having experienced symptoms that they feel may be related to COVID-19; (3) knowing anyone who tested positive for COVID-19; (4) knowing someone who died from COVID-19; and (5) job loss/reduced pay since the start of the COVID-19 pandemic.

Self-reported diagnoses. Participants were asked, “Have you ever been diagnosed with a mental health condition by a professional?” If they answered “yes”, they were presented with a list of psychiatric diagnoses from which to indicate the presence/absence of a diagnosis, including anxiety and mood disorders.

### 2.3. Procedures

Participants first provided online informed consent and completed online surveys [11]. Then, they were provided personalized feedback as an incentive for follow-up surveys. The study collected data at five time points: Time 1 (T1) from 6 April to 5 May 2020; Time 2 (T2) from 12 May to 21 June 2020; Time 3 (T3) from 25 August to 27 September 2020; Time 4 (T4) from 16 December 2020, to 19 January 2021; Time 5 (T5) from 7 April to 6 June 2021. The Institutional Review Board at the University of Pennsylvania approved all study procedures.

### 2.4. Data Analytic Plan

Participants’ engagement over time is presented in Table 1. A total of 347 participants had data on all measures at all five time points. We compared whether predisposed variables could predict whether participants completed surveys at each time point using linear regressions. As a result, age (at T3–T5, all *ps* < 0.001), gender (T2–T4, all *ps* < 0.001), and race (Black at T2 and Asian at T3 and T5, all *ps* < 0.05) were significant predictors of survey completion and were, therefore, included as covariates in sensitivity analyses for the mixed effects models, alongside a variable reflecting country (1 = living in Israel; 0 = not living in Israel). 

As an extension study of our prior investigation, we followed the same data analytic procedures as in our prior report [10]. In summary, we conducted: (1) cross-sectional multiple logistic regression analyses to predict the severity of insomnia (ISI) based on COVID-19-related exposure worries. (2) Mixed effects multilevel models (observations nested within participants) with a random intercept and slope. (3) A series of cross-lagged panel analyses to directly test the directionality between worry about COVID-19 (using sum scores of the six items to reduce the number of analyses) and ISI severity (Figure Following established procedures for this evaluation [13,14,15,16], we tested five models and the differences in their model fit to evaluate the directionality between the two variables. Detailed descriptions and rationales for each analysis can be seen in our prior report [10]. 

## 3. Results

### 3.1. Cross-Sectional Multivariable Results

Our prior report [10] reported cross-sectional multivariable results for time points 1 through 3 (times 4 and 5 were not yet collected) and demonstrated that worry variables were consistently associated with ISI severity whereas exposure variables were not. At time 4 (8 months after baseline), a cross-sectional multivariable analysis revealed that no exposure variables and two worry variables were significantly associated with ISI severity (worry about having COVID-19, *p* < 0.05; and financial worries about COVID-19, *p* < 0.001; see Table 2). At time 5 (12 months after baseline), a cross-sectional multivariable analysis revealed that only no exposure variable and three worry variables (worries about dying from COVID-19, *p* < 0.05; worries about family contracting COVID-19, *p* < 0.01; worries about finances due to COVID-19, *p* < 0.001) were associated with ISI severity.

### 3.2. Longitudinal Analyses

According to the multilevel model (observations nested within participants) with a random intercept and slope, the interactions between time and the slope of all worry variables (see Table 3, all *p*s ≤ 0.01) were significant. Specifically, over time, those who experienced more COVID-19 worries also reported more severe insomnia severity (see Figure 1A,B, for examples, though the pattern was identical for all worry outcomes). However, a reverse multilevel model also demonstrated that there was a significant interaction between time and the slope of ISI on all the worry variables (see Table 4, all *p*s ≤ 0.01). The inclusion of demographic variables (age, gender, ethnicity, country of origin) did not alter these patterns.

### 3.3. Cross-Lagged Analyses

To further elucidate the directionality between worry and insomnia, we conducted a series of cross-lagged panel analyses (Figure 2A–D). As reported in Table 5, increased model fit from Model 2 and Model 3 relative to Model 1, suggesting that cross-lagged paths were significant and should be retained (Column 5, Table 5). Model 4 significantly improved model fit above and beyond Models 2 and 3, suggesting that it is most appropriate to model cross-lagged paths within the same model (Column 5, Table 5). Finally, constraining cross-lagged paths to be equal had significantly worsened model fit than when paths were freely estimated (column 7, Table 5). This finding suggests that the strength of the cross-lagged paths was not equivalent between the constructs. To further support this possibility, the only significant cross-lagged paths in the freely estimated model (Figure 2D) were earlier in the longitudinal model (from baseline to one month) and later in the model (from 8 months to 12 months). In the earlier path, the cross-lagged path from worry to sleep was slightly larger; this was reversed in the later path. Thus, the model was significantly improved when paths could be freely estimated, but both directions appeared to be important.

## 4. Discussion

Our international longitudinal study of about 3600 participants over one year after the COVID-19 pandemic provides further support for bidirectional associations between insomnia and worry. In our original report, which tracked participants over the first 6 months of the pandemic, we found stronger evidence that worry about COVID-19 predicted future sleep disorder symptoms (specifically insomnia) [10]. As with our original report, worries about COVID-19 were more strongly associated with insomnia severity compared to exposure to COVID-19 risk factors, consistent with our first hypothesis. Worries about the financial implications of COVID-19 had a particularly strong association with insomnia. In mixed effects models, a change in worries about COVID-19 predicted a change in insomnia, and vice versa, which was consistent with our second hypothesis. However, inconsistent with our third hypothesis, we observed a reciprocal relationship between these two constructs as the pandemic continued. These findings have important implications for understanding the natural time course of sleep disorder symptoms and worry in the context of a global catastrophe with participants recruited from across the globe.

Since the publication of our initial report, which was the first to document the longitudinal time course of worries about COVID-19 alongside insomnia severity, over 40 articles have been written about sleep disorder symptoms during the pandemic. Our findings are consistent with one report that demonstrated a bidirectional, longitudinal relationship between insomnia and depression (worries were not included) over the course of one year in the pandemic [17]. Thus, it is not totally surprising that our findings were inconsistent with our third hypothesis (which was based on our prior report that observed effects in the acute phase of the pandemic). Three network analysis studies found that sleep disorder symptoms [18,19] and/or difficulty with relaxation [18] or chronic worry [19,20] were central to a network analysis relating insomnia, GAD, quality of life, and depression symptoms. However, these network analyses explored only cross-sectional associations among variables. Another cross-sectional study in psychiatric samples reported that pandemic-related worry and insomnia symptoms were associated with poorer well-being during the COVID-19 pandemic, but did not explore directional effects between these variables [21]. Multiple other cross-sectional studies from across the world reported associations between COVID-19-related worries and higher insomnia, but these studies did not track participants longitudinally [22,23,24,25,26,27,28]. Thus, our study was the first to explore long-term associations between worry and insomnia during the pandemic.

However, one other study found that worries about COVID-19 were not related longitudinally to insomnia over a shorter (3-month) window [29]. This prior report had a smaller sample size (*n* = 321) and included a slightly different measure of worry that comprised financial worries, health worries, and catastrophizing worries exclusively, which may account for the differences in the pattern across studies [29]. Other than this report, the literature is converging to highlight that worries about the pandemic and insomnia are closely related in the acute phase of a global disaster and become more reciprocally influential as the disaster continued unfolding.

Among patients with elevated COVID-19 anxiety, worries about health [30] and about other people [10] tend to be central to their anxiety. The association between chronic worry and sleep disorder symptoms has been well-established in the literature, and sleep disorder symptoms are a core diagnostic symptom of GAD, which is characterized by chronic worry [31]. The fact that sleep disorder symptoms are characterized as a symptom of GAD is a source of confusion for many who assume that this implies directionality wherein worry predicts insomnia. In fact, longitudinal research reliably demonstrates that in the general population, sleep disorder symptoms tend to precede the development of anxiety and depression [32]. In our report, we are not necessarily exploring associations between clinically significant levels of worry and insomnia, but instead evaluated worry as a spectrum. This difference may explain why we found evidence of bidirectionality, rather than unidirectionality, in our effects.

These findings offer important clinical implications. We demonstrated that we are able to predict insomnia and worry up to one year after the onset of a global catastrophe. These findings reveal that when an individual is reporting elevations in insomnia or worry, rather than assuming that these symptoms will resolve on their own, there may be value in offering targeted, evidence-based interventions to interrupt the cascade of future mental health effects. Cognitive behavioral therapy for insomnia (CBT-I) is an evidence-based treatment for insomnia that is associated with significant reductions in insomnia over time [33]. Therefore, individuals who report an elevated score on insomnia during a global disaster should be offered this intervention in order to reduce exacerbations in other mental health consequences at a future point (in this case, elevated worry). Likewise, CBT for GAD has strong efficacy and effectiveness for reducing chronic worry [34]. Individuals reporting significant elevations in worry may benefit from this intervention to reduce the risk of sleep disorder symptoms in the future. These findings suggest that rather than “waiting to see” whether clinically meaningful symptoms resolve, interventions should be offered when symptoms first emerge in the context of a global disaster. Telehealth interventions have the potential to reach more participants to reach more at-risk individuals while preserving the efficacy of the intervention [35].

There are a number of important limitations of this study that require consideration. First, participants were recruited as a convenience sample and may not reflect the general population as a result. Second, as is typical for online survey studies that do not offer compensation, many participants who completed the initial evaluation did not complete follow-up evaluations. In our initial report on this project, we evaluated the impact of missingness and determined that it was not significantly associated with our outcomes of interest. Third, the majority of respondents were White participants and women. Fourth, data were exclusively collected through self-report. Finally, we did not assess the presence or absence of diagnosed sleep disorders, which could impact the results. Despite these limitations, these findings contribute to the knowledge base on the longitudinal time course of worry and insomnia up to one year after a global disaster.

## 5. Conclusions

In conclusion, our study extends our prior report (wherein participants were followed for 6 months) by evaluating the longitudinal time course of worry and insomnia up to one year after the onset of a global disaster in an international sample. There was evidence for bidirectional associations between insomnia and worry, suggesting that these constructs were mutually influential throughout the COVID-19 pandemic. These findings have important implications for highlighting the need to offer interventions targeting elevated insomnia or worry early in a global disaster to prevent worsened future symptoms. Future research should evaluate the effect of interventions targeting worries and insomnia on the long-term change in these symptoms during a disaster.

## Figures and Tables

**Figure 1 ijerph-20-04568-f001:**
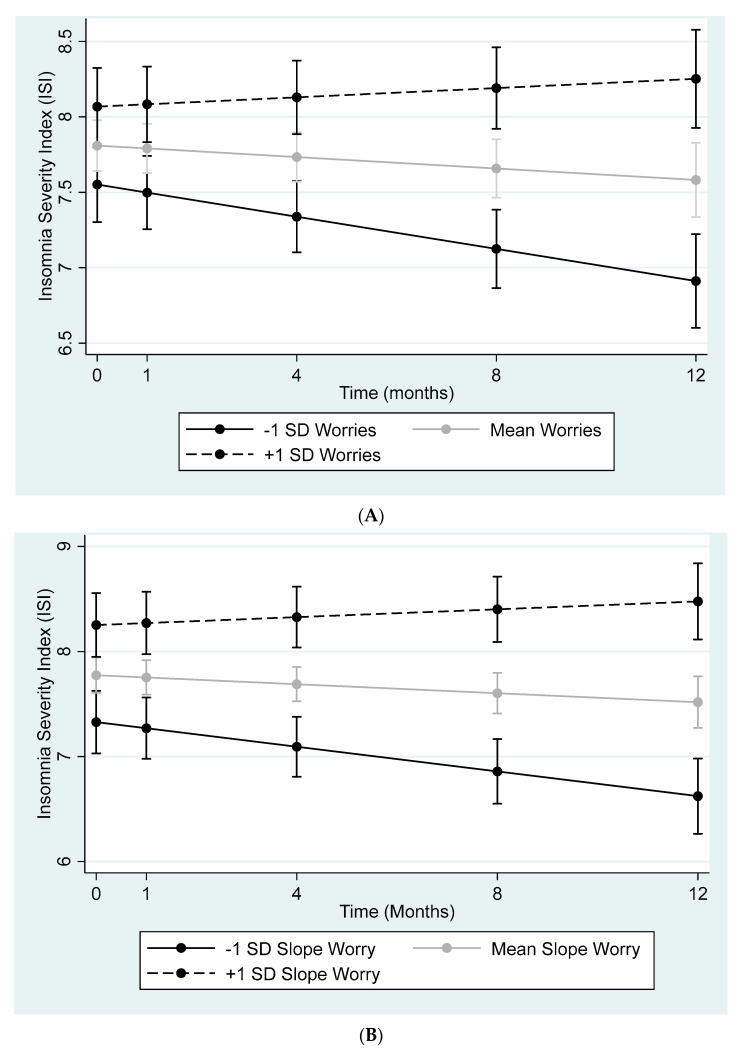
(**A**) Change in ISI predicted by change in worries about getting COVID-19. (**B**) Change in ISI predicted by change in worries about dying from COVID-19.

**Figure 2 ijerph-20-04568-f002:**
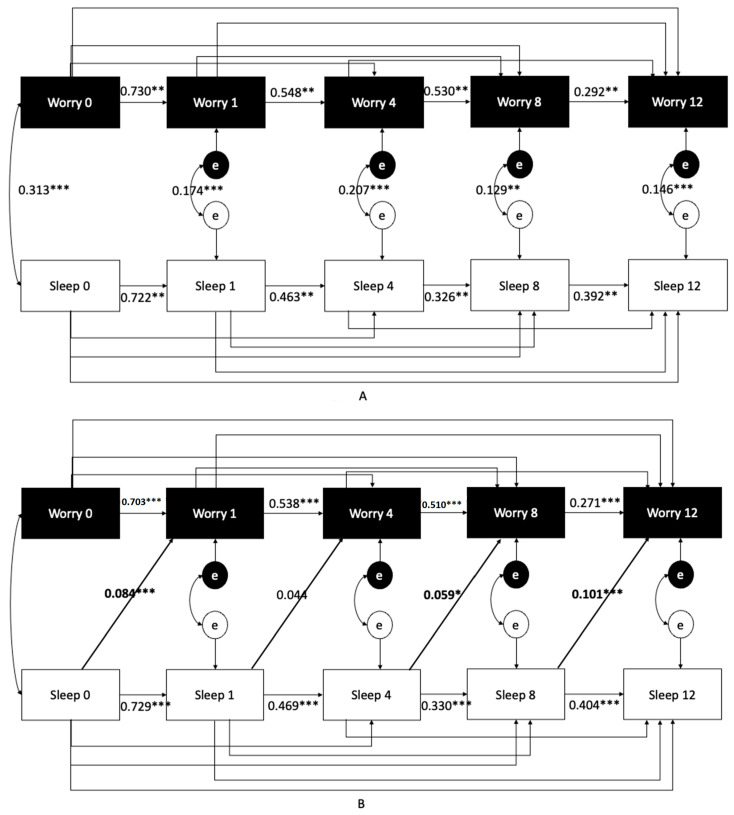
Results of a cross-lagged panel analysis to directly test directionality between Worry (calculated as a total score, a sum of all Worry variables to reduce the number of analyses) and Insomnia Severity Index severity. (**A**) Model 1 includes only the autoregressive paths. (**B**) Model 2 adds paths from one construct (sleep at T_n − 1_) to the second construct at the subsequent time-point, worry at T_n_. (**C**) Model 3 adds paths in the opposite direction to the autoregressive model: worry at T_n − 1_ predicting sleep at T_n_. (**D**) Model 4 is the full model including bidirectional paths (essentially a combination of Models 2 and 3). (**E**) Model 5 constrains cross-lagged paths to be equal at each time-point. Note: * *p* < 0.05, ** *p* < 0.01, *** *p* < 0.001.

**Table 1 ijerph-20-04568-t001:** Demographics of individuals who provided Insomnia Severity Index data at each time point.

Variable	Time 1 (*n* = 3560)	Time 2 (*n* = 1282)	Time 3 (*n* = 946)	Time 4 (*n* = 941)	Time 5 (*n* = 1170)
Age, years, mean (SD)	42.56 (14.24)	40.55 (13.47)	41.25 (14.16)	42.26 (14.59)	42.26 (14.49)
Age, years, range	8–91	16–92	8–92	11–91	11–92
Gender, *n* (%)					
Men	782 (22.0%)	230 (17.9%)	157 (16.6%)	140 (14.9%)	191 (16.3%)
Women	2766 (77.7%)	1050 (81.9%)	786 (83.3%)	799 (84.9%)	977 (83.5%)
Missing	12 (0.3%)	2 (0.2%)	1 (0.1%)	2 (0.2%)	2 (0.2%)
Race, *n* (%)					
White	3076 (86.4%)	1142 (89.1%)	855 (90.6%)	847 (90.0%)	1049 (89.7%)
Other	469 (13.2%)	135 (10.5%)	85 (9.0%)	88 (9.4%)	117 (10.0%)
Missing	15 (0.4%)	5 (0.4%)	4 (0.4%)	6 (0.6%)	4 (0.3%)
Clinical diagnosis, *n* (%)					
GAD	749 (21.0%)	321 (25.0%)	220 (23.3%)	246 (26.1%)	285 (24.4%)
No GAD	2811 (79.0%)	961 (75.0%)	724 (76.7%)	695 (73.9%)	885 (75.6%)
MDD	790 (22.2%)	313 (24.4%)	218 (23.1%)	251 (26.7%)	293 (25.0%)
No MDD	2770 (77.8%)	969 (75.6%)	726 (76.9%)	690 (73.3%)	877 (75.0%)
*Country, n (%)*					
Israel	601 (16.9%)	201 (15.7%)	144 (15.2%)	112 (11.9%)	152 (13.0%)
USA	2663 (74.8%)	1007 (78.5%)	750 (79.3%)	771 (81.9%)	931 (79.6%)
Other country/missing	386 (11.3%)	74 (5.8%)	52 (5.5%)	58 (6.2%)	87 (7.4%)
Insomnia Severity Index (range: 0–28)	7.86 (5.65)	7.35 (5.27)	7.45 (5.20)	7.83 (5.63)	7.40 (5.62)
**COVID-19-related worries (range: 0–4)**					
Worries—getting COVID-19	1.87 (1.03)	1.64 (0.94)	1.72 (0.92)	1.98 (1.01)	1.33 (0.97)
Worries—dying from COVID-19	1.18 (1.11)	0.97 (0.98)	0.99 (0.95)	1.17 (1.04)	0.82 (0.96)
Worries—family getting COVID-19	2.53 (1.07)	2.35 (1.04)	2.40 (1.01)	2.65 (1.04)	2.02 (1.08)
Worries—infecting others w/COVID-19	2.17 (1.16)	2.07 (1.12)	2.07 (1.15)	2.35 (1.19)	1.73 (1.16)
Worries—having COVID-19	0.90 (1.00)	0.65 (0.85)	0.71 (0.86)	0.87 (0.99)	0.44 (0.74)
Worries about finances due to COVID-19	1.38 (1.23)	1.26 (1.15)	1.20 (1.12)	1.10 (1.13)	0.81 (1.02)
**COVID-19-related exposures (range: 0–1)**					
Being tested for COVID-19	0.10 (0.30)	0.11 (0.32)	0.35 (0.48)	0.60 (0.49)	0.68 (0.47)
Experienced COVID-19-related symptoms	0.29 (0.46)	0.28 (0.45)	NA	NA	NA
Knowing anyone with COVID-19	0.51 (0.50)	0.59 (0.49)	0.73 (0.45)	0.86 (0.34)	0.93 (0.26)
Knowing someone died from COVID-19	0.11 (0.31)	0.15 (0.35)	0.20 (0.40)	0.24 (0.43)	0.34 (0.48)
Job loss/reduced pay since COVID-19	0.09 (0.29)	0.08 (0.28)	0.09 (0.29)	0.08 (0.28)	0.08 (0.27)

**Table 2 ijerph-20-04568-t002:** Results of cross-sectional multivariable regression results predicting ISI based on COVID-19-related exposures and COVID-19-related worries.

Time Point	B (SE)	t	*p*
*Time 4 (8 months)*			
*COVID-19-related exposures*			
Being tested for COVID-19	0.490 (0.358)	1.37	0.172
Knowing someone who tested positive for COVID-19	−0.167 (0.521)	−0.32	0.749
Knowing someone who died of COVID-19	0.221 (0.415)	0.53	0.595
Job loss/pay reduction due to COVID-19	0.063 (0.651)	0.10	0.923
*COVID-19-related worries*			
Worries about getting COVID-19	0.314 (0.264)	1.19	0.235
Worries about dying from COVID-19	0.405 (0.229)	1.77	0.078
Worries about family contracting COVID-19	0.331 (0.237)	1.40	0.163
Worries about infecting others	0.083 (0.194)	0.43	0.669
Worries about having COVID-19	0.433 (0.211)	2.06	0.040
Worries about finances due to COVID-19	1.174 (0.173)	6.77	<0.001
*Time 5 (12 months)*			
*COVID-19-related exposures*			
Being tested for COVID-19	0.508 (0.330)	1.54	0.124
Knowing someone who tested positive for COVID-19	−0.838 (0.610)	−1.37	0.170
Knowing someone who died of COVID-19	−0.327 (0.327)	−1.00	0.318
Job loss/pay reduction due to COVID-19	−0.088 (0.590)	−0.15	0.881
*COVID-19-related worries*			
Worries about getting COVID-19	0.311 (0.2375)	1.31	0.190
Worries about dying from COVID-19	0.502 (0.222)	2.26	.024
Worries about family contracting COVID-19	0.548 (0.209)	2.62	0.009
Worries about infecting others	−0.276 (0.177)	−1.56	0.118
Worries about having COVID-19	0.348 (0.236)	1.48	0.140
Worries about finances due to COVID-19	1.569 (0.1752)	8.99	<0.001

**Table 3 ijerph-20-04568-t003:** Results of longitudinal multilevel models exploring change in ISI by change in worries.

	B (95% CI)	z	*p*
*ISI*			
Time	0.028 (−0.001, 0.051)	1.95	0.051
Slope of worries—getting COVID-19	13.593 (3.663, 23.523)	2.68	0.007
Time × slope of worries—getting COVID-19	1.807 (1.080, 2.534)	4.8794	<0.001
Intercept of worries—getting COVID-19	2.251 (1.931, 2.571)	13.79	<0.001
Intercept	4.051 (3.553, 4.550)	15.94	<0.001
*ISI*			
Time	0.018 (−0.007, 0.044)	1.39	0.165
Slope of worries—dying from COVID-19	24.792 (11.403, 38.181)	3.63	<0.001
Time × slope of worries—dying from COVID-19	2.078 (1.202, 2.955)	4.65	<0.001
Intercept of worries—dying from COVID-19	1.981 (1.667, 2.296)	12.34	<0.001
Intercept	6.01 (5.726, 6.297)	41.25	<0.001
*ISI*			
Time	0.015 (−0.012, 0.043)	1.08	0.280
Slope of worries—family getting COVID-19	7.220 (−1.949, 16.390)	1.54	0.123
Time × slope of worries—family getting COVID-19	1.355 (0.655, 2.055)	3.79	<0.001
Intercept of worries—family getting COVID-19	2.083 (1.822, 2.343)	15. 68	<0.001
Intercept	2.763 (2.166, 3.360)	9.07	<0.001
*ISI*			
Time	0.046 (0.013, 0.078)	2.76	0.006
Slope of worries—infecting others w/COVID-19	−4.374 (−18.943, 10.196)	−0.59	0.556
Time × slope of worries—infect others w/COVID-19	3.026 (1.888, 4.163)	5.21	<0.001
Intercept of worries—infect others w/COVID-19	1.539 (1.298, 1.780)	12.51	<0.001
Intercept	4.403 (3.886, 4.921)	16.67	<0.001
*ISI*			
Time	0.036 (0.001, 0.071)	2.03	0.042
Slope of worries—having COVID-19	50.262 (9.368, 91.157)	2.41	0.016
Time × slope of worries—having COVID-19	2.122 (1.056, 3.187)	3.90	<0.001
Intercept of worries—having COVID-19	4.140 (2.827, 5.453)	6.18	<0.001
Intercept	5.590 (5.280, 5.901)	35.29	<0.001
*ISI*			
Time	0.032 (−0.006, 0.069)	1.66	0.096
Slope of worries—finances due to COVID-19	35.634 (24.014, 47.255)	6.01	<0.001
Time × slope of worries—finances due to COVID-19	1.216 (0.493, 1.939)	3.30	0.001
Intercept of worries—finances due to COVID-19	2.551 (2.236, 2.866)	15.88	<0.001
Intercept	5.810 (5.466, 6.154)	33.11	<0.001

**Table 4 ijerph-20-04568-t004:** Results of longitudinal multilevel models exploring change in Worries by change in ISI.

	B (95% CI)	z	*p*
*Worries about getting COVID-19*			
Time	−0.026 (−0.030, −0.022)	−12.07	<0.001
ISI Slope	−0.314 (−0.774, 0.146)	−1.34	0.181
Time × ISI Slope	0.109 (0.064, 0.155)	4.76	<0.001
ISI Intercept	0.052 (0.045, 0.059)	14.35	<0.001
Intercept	1.446 (1.385, 1.507)	46.58	<0.001
*Worries about dying of COVID-19*			
Time	−0.018 (−0.021, −0.014)	−9.26	<0.001
ISI Slope	−0.249 (−0.754, 0.256)	−0.97	0.334
Time × ISI Slope	0.100 (0.061, 0.139)	4.99	<0.001
ISI Intercept	0.056 (0.047, 0.063)	13.83	<0.001
Intercept	0.728 (0.661, 0.795)	21.38	<0.001
*Worries about family contracting COVID-19*			
Time	−0.026 (−0.030, −0.021)	−11.41	<0.001
ISI Slope	−0.177 (−0.656, 0.303)	−0.72	0.471
Time × ISI Slope	0.085 (0.040, 0.131)	3.67	<0.001
ISI Intercept	0.062 (0.054, 0.069)	16.04	<0.001
Intercept	2.047 (1.983, 2.111)	62.43	<0.001
*Worries about infecting others*			
Time	−0.020 (−0.025, −0.015)	−8.41	<0.001
ISI Slope	−0.573 (−1.096, −0.049)	−2.14	0.032
Time × ISI Slope	0.124 (0.076, 0.173)	5.07	<0.001
ISI Intercept	0.053 (0.044, 0.061)	12.35	<0.001
Intercept	1.745 (1.675, 1.817)	48.16	<0.001
*Worries about having COVID-19*			
Time	−0.025 (−0.029, −0.021)	−12.86	<0.001
ISI Slope	−0.655 (−1.078, −0.232)	−3.03	0.002
Time × ISI Slope	0.091 (0.050, 0.132)	4.34	<0.001
ISI Intercept	0.048 (0.041, 0.054)	14.86	<0.001
Intercept	0.485 (0.430, 0.540)	17.39	<0.001
*Worries about finances during COVID-19*			
Time	−0.041 (−0.045, −0.036)	−18.40	<0.001
ISI Slope	0.137 (−0.413, 0.687)	0.49	0.625
Time × ISI Slope	0.091 (0.044, 0.137)	3.84	<0.001
ISI Intercept	0.077 (0.069, 0.086)	18.26	<0.001
Intercept	0.791 (0.719, 0.862)	21.68	<0.001

**Table 5 ijerph-20-04568-t005:** Cross-lagged panel analysis to examine potential bidirectionality between COVID-19 worries and insomnia.

Model	df	χ^2^	Scale Correlation Factor	Satorra–Bentler Scaled χ^2^ from Base Model	Satorra–Bentler Scaled χ^2^ from Full Model	Constrained vs. Freely Estimated Model Difference (Using Correlation Matrix)	CFI	TLI	AIC	AIC Δ from Base Model	AIC Δ from Full Model	SRMR	RMSEA
Baseline Model	20	84.765	1.187	-	70.53 ***	-	0.987	0.97	112,827.7	-	65.7	0.063	0.021
Sleep → Worry (Model 2)	16	41.544	1.184	42.89 ***	27.08 ***	-	0.995	0.985	112,784.3	−43.4	22.3	0.026	0.015
Worry → Sleep (Model 3)	16	53.413	1.2025	32.34 ***	38.02 ***	-	0.992	0.979	112,799.3	−28.4	37.3	0.04	0.018
Full Model	12	15.683	1.2061	70.53 ***	-	-	0.999	0.997	112,762	−65.7	-	0.01	0.006
Full Model with Constraint	16	19.115	1.188	65.86 ***	3.35	50.97 ***	0.999	0.998	112,758.5	−69.2	−3.5	0.011	0.005

Note: *** indicates *p* < 0.01.

## Data Availability

Available upon request.

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
