# Peer review of "COVID-19 Worries and Insomnia: A Follow-Up Study"

_ijerph, 2023, doi:10.3390/ijerph20054568_

Round 1

Reviewer 1 Report

Kindly revise the writing style and the Result section again to make sure the precision of the results. The other parts are okay and I am happy to accept the paper after improving the result section writing. 

Author Response

Reviewer 1:

Kindly revise the writing style and the Result section again to make sure the precision of the results. The other parts are okay and I am happy to accept the paper after improving the result section writing. 

Thank you for the feedback. In addition to addressing comments that other reviewers has on our results section, we closely reviewed the results section to ensure readability, and hope that you would agree that our revision attempt improved the result section writing.

Reviewer 2 Report

Dear Editor and Authors,

Thank you for the opportunity to review the manuscript entitled COVID-19 worries and insomnia: A follow-up study. The purpose of the present study was to evaluate the bidirectional associations between insomnia and worry over one year of the pandemic. The self-reported surveys were completed at five time points including worries about the pandemic, exposure to virus risk-factors, and the Insomnia Severity Index. Results revealed that rates of insomnia reduced over time and change in worries about COVID-19 predicted change in insomnia, and vice versa. This manuscript was an interesting read and made an important contribution to research in the context of COVID-19 pandemic. It also seems to be a good fit for this journal. However, some critical points need to be considered in this study. I offer several suggestions for the authors' consideration below

·       Introduction

Generally, the introduction seems relatively well-written.

Data for the current study were collected in the United States and Israel and authors explored the temporal direction of worry and insomnia during the acute phase. I would suggest that provide more details about rates of insomnia, pre-pandemic and during the pandemic rates, in these countries. Importantly, the progression of the pandemic and pandemic-related measures were differed in these countries at five time points of data collection which might be affected on COVID-19 worry. I would suggest to provide further information about pandemic statistics and measures focusing the US and Israel over one year of data collection.

·       Method

Provide further information about inclusion criteria and exclusion criteria of the participants.

Participants were recruited as a convenience sample in the current study but authors didn’t provide details. I suggest that the authors include future information about the sample.

I would suggest that authors add more details about how participants were asked about clinical diagnosis (Table 1).

I would suggest that authors add more information about how authors analyzed data in order to study predisposed variables predicting participants' engagement over time (Line 103-108).

Provide more information about psychometric properties of scales.

·       Results

In the results section, I would suggest the authors add more analysis related to rates of insomnia and worries about the pandemic as function of country (US vs Israel).

I would suggest that authors add one more table to show descriptive information of worries about the pandemic, exposure to virus risk-factors, and the Insomnia Severity Index at five time points.

·       Discussion

Although findings are interesting, the authors actually need to add more details in order to discuss their main results in the discussion. Discussion should be revised with more details. Moreover, COVID-19-related exposures variables were not associated with ISI severity. These findings need to be discussed.

In the current study, the finding showed significant reductions in insomnia over time. This finding needs to be discussed.

Pervious study showed that COVID-19 worries were associated with mental health problems including insomnia and one’s personal finances was an important predictor. In the current study, worries about finances due to COVID-19 was also strongest predictor compared with other worry variables. This finding also needs to be discussed.

The authors seem to have a good understanding of the limitations of their findings. I suggest that the authors include future studies in order to further stimulate additional research.

Lack of baseline data. That is the major limitation according to me and needs to be discussed further.

·       Minor comments

Line 85: ”Likert” appeared two times. Delete one of them

Line 96: add 2020 to the sentence “Time 2 (T2) from May 12 to June 21”

I hope that these comments and suggestions are helpful for authors and that they can be used to improve this interesting work.

Author Response

Reviewer 2:

Dear Editor and Authors,

Thank you for the opportunity to review the manuscript entitled COVID-19 worries and insomnia: A follow-up study. The purpose of the present study was to evaluate the bidirectional associations between insomnia and worry over one year of the pandemic. The self-reported surveys were completed at five time points including worries about the pandemic, exposure to virus risk-factors, and the Insomnia Severity Index. Results revealed that rates of insomnia reduced over time and change in worries about COVID-19 predicted change in insomnia, and vice versa. This manuscript was an interesting read and made an important contribution to research in the context of COVID-19 pandemic. It also seems to be a good fit for this journal. However, some critical points need to be considered in this study. I offer several suggestions for the authors' consideration below

  • Introduction
  • Generally, the introduction seems relatively well-written.

Thank you for the encouraging feedback.

  • Data for the current study were collected in the United States and Israeland authors explored the temporal direction of worry and insomnia during the acute phase. I would suggest that provide more details about rates of insomnia, pre-pandemic and during the pandemic rates, in these countries. Importantly, the progression of the pandemic and pandemic-related measures were differed in these countries at five time points of data collection which might be affected on COVID-19 worry. I would suggest to provide further information about pandemic statistics and measures focusing the US and Israel over one year of data collection.

This is an excellent point. We added prevalence rates in the introduction.

  • Method
  • Provide further information about inclusion criteria and exclusion criteria of the participants.

Thank you for the suggestion. We added an additional section of participants to provide more information. For example, we explained that there is no inclusion or exclusion criteria and explained how participants were recruited. We also directed readers to other report for more information, including the self-reported history of diagnosis.

  • Participants were recruited as a convenience sample in the current study but authors didn’t provide details. I suggest that the authors includefuture information about the sample.

Thank you for the important point. As addressed above, we added a section to explain and provide more details.

  • I would suggest that authors add more details about how participants were asked about clinical diagnosis (Table 1).

Thank you for raising this. To address this, we added the following to the methods:

“Self-reported diagnoses. Participants were asked “Have you ever been diagnosed with a mental health condition by a professional?” If they answered “yes”, they were presented with a list of psychiatric diagnosis from which to indicate the presence/absence of a diagnosis, including anxiety and mood disorders.”

  • I would suggest that authors add more information about how authors analyzed data in order to study predisposed variables predicting participants' engagement over time (Line 103-108).

Thank you for the suggestion. We now pointed out that covariates were added to the mixed effects model results in sensitivity analyses.

  • Provide more information about psychometric properties of scales.

Thank you for raising this. The measures for the study were deployed early in the pandemic and were not subjected to psychometric evaluations before they were released. However, psychometric performance is reported for the Insomnia Severity Index.

  • Results
    • In the results section, I would suggest the authors add more analysis related to rates of insomnia and worries about the pandemic as function of country (US vs Israel).

Thank you for this suggestion. Based on this suggestion, we added in country as a covariate in the multilevel analyses, and this did not alter any of the results.

  • I would suggest that authors add one more table to show descriptive information of worries about the pandemic, exposure to virus risk-factors, and the Insomnia Severity Index at five time points.

Thanks for the suggestion. We added the descriptive information of ISI, worry, exposure variables to the end of Table 1 (M, SD, and possible Range) at each wave.

  • Discussion
    • Although findings are interesting, the authors actually need to add more details in order to discuss their main results in the discussion. Discussion should be revised with more details. Moreover, COVID-19-related exposures variables were not associated with ISI severity. These findings need to be discussed.
  • Thank you for raising this. Our discussion in organized as follows:
  • 1) Paragraph 1: An overview of the key findings and the “message” of the paper, including the lack of significant effects of exposure variables
  • 2) Paragraph 2: Discussion of our key findings as they related to our hypotheses
  • 3) Paragraph 3: Differences between our findings and that of other reports in the literature
  • 4) Paragraph 4: Connection between our findings to other literature
  • 5) Paragraph 5: Clinical implications of our findings
  • 6) Paragraph 6: Limitations
  • 7) Paragraph 7: Summary of our findings and overview
    • In the current study, the finding showed significant reductions in insomnia over time. This finding needs to be discussed.

Of note, when reviewing Table 3 (which has been updated to reflect the changes suggested by you and other reviewers), in all instances the main effect of Time was not significant. We have clarified this in to reduce confusion for the readers.

  • Pervious study showed that COVID-19 worries were associated with mental health problems including insomnia and one’s personal finances was an important predictor. In the current study, worries about finances due to COVID-19 was also strongest predictor compared with other worry variables. This finding also needs to be discussed.

This is an important point. To address this, we added the following to the discussion:

“Worries about the financial implications of COVID-19 had a particularly strong association with insomnia.”

  • The authors seem to have a good understanding of the limitations of their findings. I suggest that the authors includefuture studies in order to further stimulate additional research.

Thank you for the feedback. To address this, we added the following:

“Future research should evaluate the effect of interventions targeting worries and insomnia on the long-term change in these symptoms during a disaster.”

  • Lack of baseline data. That is the major limitation according to me and needs to be discussed further.

Thank you for raising this important point. We added data in Table 1 to reflect scores at baseline and each of the subsequent time-points.

  • Minor comments
  • Line 85: ”Likert” appeared two times. Delete one of them

Thank you for alerting us. We edited this accordingly.

  • Line 96: add 2020 to the sentence “Time 2 (T2) from May 12 to June 21”

Thanks for catching that. We have edited this accordingly.

I hope that these comments and suggestions are helpful for authors and that they can be used to improve this interesting work.

Thank you for your helpful feedback

Reviewer 3 Report

The current article (Covid-19 concerns and insomnia: A follow-up study) assesses the impact of the current pandemic on insomnia symptoms.

The study is one of many developed as an online survey. However, it properly addresses the limitations of the chosen methodology and may add some data on the populations of the US and Israel.

There are, however, some important considerations that need to be made:

  1. It is important that the introduction be concise, but there are numerous studies on the prevalence of insomnia during the COVID period, so I think more studies need to be presented in the introduction.
  2. Methodology:
  • More details are needed on the procedure chosen.
    • Online platforms are used.
    • Response rate.
    • How was the questionnaire distributed?
    • It is not clear whether those who filled out the questionnaire five times (Q1-T5) are the same people or different people.
      • If they are the same subjects, it is not clear from the methodology how the statistical analysis was conducted and whether the values presented have a role in longitudinal evaluation or are the isolated results of several cross-sectional studies.
      • If they are not the same subjects, you cannot present the study as a follow-up one.
    • More data on statistical analysis is needed to be described.
    • Is the database accessible online? I would have wanted to run some of the statistical tests again.
    • Did the subjects in Israel fill in the questionnaire in English? If so, how did you check their understanding of the questions?
    • Are there any criteria for inclusion or exclusion?
    • The batch in your first study (reference no. 11; T1-T3) and the batch in the current study differ slightly.

  1. Results
    • Lines 124–125: It is difficult to understand the message, and I think it is important to present initially what was correlated.
    • Lines 128–130: As in the previous point, I think a simpler or clearer expression would help.
    • Fig. 1a and Fig. 1b are redundant in terms of T4. Figure 1B is sufficient. On the OX axis, what is presented?
    • I believe the results in tables 3 and 4 could be presented more clearly.
    • Lines 147–162: The models discussed must be defined.
  2. Discussion
    • There are cultural and social differences between the U.S. and Israel. Why did you choose to present the results together?
      • If there are differences, they must be specified in the results.
      • Even if there are no significant differences, this should be discussed in the discussion section.

Author Response

Reviewer 3:

The current article (Covid-19 concerns and insomnia: A follow-up study) assesses the impact of the current pandemic on insomnia symptoms.

The study is one of many developed as an online survey. However, it properly addresses the limitations of the chosen methodology and may add some data on the populations of the US and Israel.

There are, however, some important considerations that need to be made:

  1. It is important that the introduction be concise, but there are numerous studies on the prevalence of insomnia during the COVID period, so I think more studies need to be presented in the introduction.

Thank you for this point, and we agree. Accordingly, we have added prevalence information to the introduction.

  1. Methodology:
  • More details are needed on the procedure chosen.
    • Online platforms are used.

Thank you – we now added additional information about this in the Participants section. Specifically, we pointed out that “participants completed online surveys in English or Hebrew through a crowdsourcing website”. We also provided the reference of the source paper (https://doi.org/10.1038/s41398-020-00982-4 ) for readers interested in details of the platform.

    • Response rate.

Thank you –7,690 individuals started our questionnaire and among them 3560 individuals (46.3%) provided data on the Insomnia Severity Index at Time 1. We added this information to the Participants section.

    • How was the questionnaire distributed?

Thank you – we now added additional information about this in the Participants section.

    • It is not clear whether those who filled out the questionnaire five times (Q1-T5) are the same people or different people.
      • If they are the same subjects, it is not clear from the methodology how the statistical analysis was conducted and whether the values presented have a role in longitudinal evaluation or are the isolated results of several cross-sectional studies.

Thank you for raising this important point. In the prior submission, we included all available participants who provided any data (including folks who provided T1 data after the original data were published). We realize that this is confusing, and now restricted the analyses to include only participants who were included in the original report. The restriction did not change patterns of our results.

      • If they are not the same subjects, you cannot present the study as a follow-up one.

Thank you we have now clarified this above.

    • More data on statistical analysis is needed to be described.

Thank you for raising this. We added additional details to the data analysis section and throughout the results to clarify.

    • Is the database accessible online? I would have wanted to run some of the statistical tests again.

We do not have IRB permission to post the database online but are happy to share the database with the reviewer (or other future collaborators) upon request.

    • Did the subjects in Israel fill in the questionnaire in English? If so, how did you check their understanding of the questions?

Thank you for raising this. Participants in Israel completed the survey in Hebrew, which we now clarified.

    • Are there any criteria for inclusion or exclusion?

Thank you for requesting this clarification. We have now added that there are no inclusion/exclusion criteria for the study.

    • The batch in your first study (reference no. 11; T1-T3) and the batch in the current study differ slightly.

Thanks for catching that. Since our initial investigation, some additional participants were recruited, which explained the discrepancies. We re-ran our analyses using data collected from those with original data, to match our sample in the prior study. With our new analyses, the demographics (e.g., age, race, gender) on T1-T3 matched our prior report.

  1. Results
    • Lines 124–125: It is difficult to understand the message, and I think it is important to present initially what was correlated.

We agree with this point and have edited this line accordingly.

    • Lines 128–130: As in the previous point, I think a simpler or clearer expression would help.

We agree and have clarified this accordingly.

    • Fig. 1a and Fig. 1b are redundant in terms of T4. Figure 1B is sufficient. On the OX axis, what is presented?

Thank you for raising this. We now specify that the X axis is Time (Months). We would be happy to remove one or both figures but have retained them as it may facilitate understanding the of the nature of the effects over time.

    • I believe the results in tables 3 and 4 could be presented more clearly.

Thank you – we have revised the results in Tables 3 and 4 to promote clarity.

    • Lines 147–162: The models discussed must be defined.

We have clarified that these models are cross-lagged panel models and include fit information in Table 5 to clarify.

  1. Discussion
    • There are cultural and social differences between the U.S. and Israel. Why did you choose to present the results together?

Thank you for raising this – we added country as a covariate in the analyses for a sensitivity check, and this not alter the pattern of results.

      • If there are differences, they must be specified in the results.

Thank you for raising this, which we addressed above.

      • Even if there are no significant differences, this should be discussed in the discussion section.

Thank you – in several places throughout the discussion section we now remind readers that this was an international sample.

Round 2

Reviewer 2 Report

Thank you for the revision of the paper. It is much clearer now and the authors have addressed all the comments given. 

A minor comment: 

Please add participants' age range to Table 1. 

Author Response

Thank you - we have added the age range to Table 1.
